# Fatty Acids Profile, *Trans* Isomers, and Lipid Quality Indices in Smoked and Unsmoked Cheeses and Cheese-Like Products

**DOI:** 10.3390/ijerph17010071

**Published:** 2019-12-20

**Authors:** Beata Paszczyk, Magdalena Polak-Śliwińska, Joanna Łuczyńska

**Affiliations:** Faculty of Food Sciences, University of Warmia and Mazury in Olsztyn, 10-719 Olsztyn, Poland; m.polak@uwm.edu.pl (M.P.-Ś.); jlucz@uwm.edu.pl (J.Ł.)

**Keywords:** cheese, cheese-like products, fatty acids, *trans* isomers, lipid quality indices

## Abstract

The purpose of this study was to evaluate the fatty acid composition, including *trans* C18:1 and C18:2 isomers and the content of conjugated linoleic acid *cis*9*trans*11 C18:2 (CLA), in commercial smoked and unsmoked cheeses and cheese-like products available on the Polish market as well as to compare lipid quality indices in these products. The composition of fatty acids was determined with the gas chromatography method. The conducted study demonstrated that smoked and unsmoked cheeses as well as smoked and unsmoked cheese-like products were characterized by various contents of fatty acids and various lipid quality indices. The smoked and the unsmoked cheeses had significantly higher (*p* < 0.05) contents of saturated fatty acids (SFA), short-chain fatty acids (SCFA), and branched-chain fatty acids (BCFA) than the smoked and the unsmoked cheese-like products. The monounsaturated fatty acids (MUFA) and the polyunsaturated fatty acids (PUFA) contents were the highest in unsmoked cheese-like products (39.29 ± 1.49% and 9.13 ± 0.33%, respectively). In smoked and unsmoked cheeses, MUFA were above 24% and PUFA were above 2.4%. The total content of *trans* C18:1 isomers was significantly higher (*p* < 0.05) in the cheeses, but in the group of these isomers, *trans*10 + *trans*11 isomers were dominant. High levels of *trans*6–*trans*9 isomers (up to 2.92% of total fatty acid) were found in some of the samples of unsmoked cheese-like products, while their content in cheeses was lower. The lipid quality indices in cheeses and cheese-like products were varied. The smoked and the unsmoked cheeses were characterized by significantly higher (*p* < 0.05) values of the index of thrombogenicity (TI) and atherogenicity (AI) indices and significantly lower (*p* < 0.05) values of the hypocholesterolemic/hypercholesterolemic (HH) ratio.

## 1. Introduction

All natural fats and oils are a combination of monounsaturated, polyunsaturated, and saturated fatty acids. A characteristic feature of milk fat is the presence of saturated fatty acids (SFA) as well as a high content of short-chain fatty acids (SCFA). Typical milk fat from dairy cows contains about 70% of SFA, 25% of monounsaturated fatty acids (MUFA), and about 5% of polyunsaturated fatty acids (PUFA) [1]. Of the saturated fatty acids, about 10% are short-chain fatty acids (C4:0–C10:0) [2]. Studies on fatty acid effects on human health indicate that only a few individual fatty acids are responsible for the negative consequences on consumer health [3,4]. According to Ulbricht and Southgate, acids such as lauric (C12:0), myristic (C14:0), and palmitic (C16:0) acid are highly related to an increased risk of atherosclerosis, obesity, and coronary heart diseases [5]. Studies of other authors assessing the impact of saturated fat intake on metabolic health are inconclusive [6] and indicate that the physiological impact of dietary SFA depends not only on dietary source and food matrix but also on SFA type and composition [7,8]. In milk fat, SFAs are the predominant class of fatty acids (FA), but these fats are unique as they also comprise a wide and complex variety of SFA, including short-chain fatty acids, medium and long-chain fatty acids, as well as odd and branched-chain fatty acids [2,9]. Studies by Kim and Je [10] have shown that the consumption of dairy products may reduce the risk of metabolic diseases. Unsaturated fatty acids (UFAs) are usually called “healthy fats”, especially for their impact on the level of cholesterol in blood [11,12]. PUFAs decrease cholesterol content more strongly than monounsaturated fatty acids (MUFAs) [13]. *n*-3 PUFA prevents heart disease and improves immune response. Oleic acid (*cis*9 C18:1) and linolenic acid (*cis*9*cis*12*cis*15 C18:3) have anti-cancer and anti-atherogenic properties [11,13]. *n*-6 PUFA improves sensitivity to insulin and thus reduces the incidence of type 2 diabetes [12].

*Trans* fatty acids (TFAs) are unsaturated fatty acids that contain at least one double bond in the *trans* configuration. *Trans* fats are found in foods originating from ruminant animals and in foods containing partially hydrogenated vegetable oils (PHVO). The levels of natural *trans* isomers in milk and meat ruminant animals can comprise up to 6% of the total content of fatty acids, and industrial *trans* fat levels can comprise up to 60% of the total content of fatty acids. The TFA content of partially hydrogenated vegetable oils (PHVO) depends on the variables of the hydrogenation process, e.g., time, catalyst, temperature, hydrogen pressure, and types and proportions of oils and composition of fatty acids. The primary dietary TFAs are vaccenic (*trans*11 C18:1) and elaidic acid (*trans*9 C18:1). Vaccenic acid is the major ruminant TFA, whereas elaidic acid is the main TFA isomer in industrial hydrogenation [14]. According to literature data, some fatty acids with *trans* configuration have an adverse effect on human health [15,16]. Epidemiological studies have shown a direct association between the intake of TFA and the risk of coronary heart disease (CHD), primarily accounted for by industrially produced TFA [17,18,19,20,21].

In its composition, milk fat contains bioactive components that have an impact on human health such as branched-chain fatty acids (BCFA), *trans* vaccenic acid (*trans* 11 C18:1), and conjugated linoleic acid *cis*9*trans*11 C18:2 (CLA), which are characteristic constituents of milk fat. Branched-chain fatty acids (BCFA, *iso*- and *anteiso*) arise in the rumen, where they are synthesized de novo or metabolized from phytol by rumen microorganisms. BCFA represents about 2% of fatty acids in cow’s milk fat [8]. Conjugated linoleic acid (CLA) is a mixture of geometrical and positional isomers of linoleic acid LA (*cis*9,*cis1*2 C18:2), which contains a conjugated double bond system and involves a double bond at positions 8 and 10, 9 and 11, and 10 and 12 or 11 and 13 [22]. Conjugated linoleic acid *cis*9*trans*11 C18:2 (CLA) has health-positive properties including, e.g., anti-carcinogenic, anti-atherosclerotic, anti-oxidative, and anti-inflammatory effects [23,24,25,26,27,28]. Experimental studies have shown that t*rans*-vaccenic acid, the main *trans* C18:1 isomer in milk fat, has anti-cancer and anti-atherogenic effects [29].

Cheeses are a significant component of the human diet. The assortment of cheeses on the Polish market is very wide. Cheese is a significant source of fat in the human diet and contains a high variety of fatty acids. Cheese fat currently suffers from an adverse nutritional image, largely due to a perceived association of saturated and *trans* fatty acids with cardiovascular diseases. Nevertheless, despite the presence of considerable amounts of such fatty acids, there is no clear evidence relating cheese consumption to any disease. Furthermore, cheese contains other fatty acids, e.g., conjugated linoleic acid and oleic acid, which have the potential to improve long-term health. Cheese-like products (or cheese analogues) are usually defined as products made by blending individual components, including non-dairy fats or proteins, to produce a cheese-like product to meet specific requirements. They are being increasingly used due to their cost-effectiveness, which is attributable to the simplicity of their manufacture and the replacement of selected milk ingredients by cheaper vegetable products [30,31]. 

The smoking of dairy products such as cheese is one of the oldest methods used in the preservation and the production of flavored food. This process provokes important modifications in food composition and sensory attributes. Smoking not only extends the shelf life of food via the effect of dehydration and antioxidant action of the smoke compound, which has the effect on microbial activity, smoking also gives special color and flavor to the food. The smoke can be applied to cheese as natural smoke or as a flavoring smoke. The wood, the smoke generation method, and the procedure have a decisive influence on cheese color, smell, taste, and texture as well as on its oxidative stability, microbial growth, and safety. The smoking process significantly enriches the cheese with volatile compounds belonging to a wide variety of chemical families such as: aldehydes (vanillin, furfural, benzaldehyde); furanmethanol; cyclic ketones related to nutty, burnt, coffee, caramel notes [32]; aromatic ketones; furan and pyran derivatives that give pleasant flavors; and nitrogen derivatives, mainly phenol, guaiacol, and syringol derivatives [33,34]. Smoked cheese is appreciated by consumers due to its sensorial properties. However, with the smoking process, there is a risk of the formation of undesired substances such as lipids peroxidation products, which have a significantly important impact on human health. There is a lack of literature data on the impact of smoking on the fatty acids profile in the smoked dairy products. The research by El-Tahra et al. [35] showed that admixing smoke liquid or powder to goat’s milk reduced the content of saturated fatty acids and increased the content unsaturated fatty acids of Domiatti cheese. The contents of short and medium-chain fatty acids were lower, and long-chain fatty acids were higher in smoked cheese as compared with control samples.

Therefore, the purpose of this study was to evaluate the fatty acid composition, including *trans* C18:1 and C18:2 isomers and content of conjugated linoleic acid *cis*9*trans*11 C18:2 (CLA) in commercial smoked and unsmoked cow cheeses and cheese-like products available on the Polish market and to compare lipid quality indices in these products.

## 2. Materials and Methods 

### 2.1. Materials

The experimental materials were commercial rennet ripening smoked and unsmoked cheese produced with cow milk and commercial smoked and unsmoked cheese-like products available on the Polish market (Table 1). The products originated from various Polish producers and were purchased in stores in Olsztyn in the period from April to May 2018. Each sample was analyzed in duplicate.

### 2.2. Analytical Methods

#### 2.2.1. Fat Extraction

The lipids were extracted according to the modified Folch’s procedure [36]. The studied material was crushed and mixed. Approximately 3 g of samples (0.01 g) was homogenized (IKA Ultra-Turrax^®^T18 digital) for 1 min with 30 mL of methanol. Next, 30 mL chloroform was added, and the procedure was continued for 2 min. The prepared mixture was filtered to a 250 mL glass cylinder. The solid residue was mixed in 60 mL chloroform: methanol (2:1 *v*/*v*) and homogenized again for 3 min. The mixture was transferred to the same cylinder. Next, 0.88% sodium chloride in water was added to the total filtrate (in the amount constituting 1/4 volume of filtrate), then shaken and left overnight. The upper layer was removed using a water pump. A water: methanol mixture (1:1 *v*/*v*) was added to the lower layer. The washing procedure was repeated. The remaining layer was filtered through anhydrous (VI) sodium sulfate and distilled for complete evaporation of the solvent.

#### 2.2.2. Preparation of Fatty Acid Methyl Esters

Fatty acid methyl esters were prepared according to IDF method (ISO 15884:2002) [37]. To do so, 50 mg of extracted fat was placed inside a sealed ampule, then 2.5 mL hexane was added to dissolve it. The container was shaken vigorously until the fat was completely dissolved. Next, 0.1 mL 2M methanolic KOH solution was added. The ampule was shaken for 1 min and then left for 5 more minutes. After that time, 0.25 g of NaHSO_4_ × H_2_O was added and spun for 3 min in a separator (approximately 1000 spins/min). The top layer of prepared methyl esters was taken for chromatographic analysis.

#### 2.2.3. Gas Chromatography (GC) Analysis

The composition of fatty acids was determined applying the method of gas chromatography with the help of Hewlett Packard 6890 GC System (Műnster, Germany) with a flame ionization detector (FID) in 100 m capillary column (produced by Chrompack, Middelburg, the Netherlands) with CP Sil 88 phase. The column diameter was 0.25 mm, and the film was 0.20 μm thick. The used separation conditions are presented in Table 2. 

Identification of fatty acids was carried out based on the comparison of their retention time with the retention time of methyl esters of fatty acids of reference milk fat (BCR Reference Materials) of CRM 164 symbol. Examples of chromatograms are shown in Figure 1. The positional *trans* isomers of C18:1 were identified using the standards of methyl esters of these isomers (*trans*6, Supelco and *trans*9 and *trans*11, Sigma-Aldrich, St. Louis, MO, USA), whereas the *trans* isomers of C18:2 acid (*cis,trans* and *trans,cis*) were identified with the use of a mixture of standards of C18:2 isomers (Supelco). The *cis*9*trans*11 CLA isomer was identified using a mixture of CLA methyl esters (Sigma-Aldrich). The proportions of the individual acids were calculated by the ratio of their peak area to the total area of all identified acids (% mass fraction).

### 2.3. The Lipid Quality Indices Were Calculated from the Fatty Acids Composition Using the Following Formulae

#### 2.3.1. Index of Atherogenicity (AI)

The index of atherogenicity (AI) indicates the relationship between the sum of the main saturated fatty acids and that of the main classes of unsaturated fatty acids. The clotting activity of platelets and their aggregation to a thrombus is closely related to the intake of long-chain SFA. *n*-6 PUFA are antiatherogenic (reduction of serum lipids), whereas *n*-3 PUFA are antithrombogenic (inhibiting the aggregation of platelet, diminishing the levels of cholesterol and phospholipids, thereby preventing the appearance of coronary disease) (Equation (1)) [5,38]:
AI = (C12:0 + (4 × C14:0) + C16:0)/(Σ*n*-3 PUFA + Σ*n*-6 PUFA + Σ MUFA)(1)

#### 2.3.2. Index of Thrombogenicity (TI)

The index of thrombogenicity (TI) indicates there is a tendency for clots to form in the blood vessels. This is defined as the relationship between the pro-thrombogenetic (saturated) and the anti-thrombogenetic fatty acids (MUFA, PUFA *n*-6 and PUFA *n*-3) (Equation (2)) [3,18].
TI = (C14:0 + C16:0 + C18:0)/((0.5 × C18:1) + (0.5 × other MUFA) + (0.5 × Σ*n*-6 PUFA) + (3 × Σ*n*-3 PUFA) + Σ*n*-3 PUFA/Σ*n*-6 PUFA))(2)

#### 2.3.3. Hypocholesterolemic Fatty Acids (DFA)

The nutritional quality indices of lipids were determined using the fatty acids, estimating the unsaturated fatty acids (UFA) by summing the polyunsaturated (PUFA) and the monounsaturated (MUFA), and using the desirable saturated fat acids (C18:0) (Equation (3)) [38]. DFA = UFA + C18:0(3)

#### 2.3.4. Hypercholesterolemic Fatty Acids (OFA) (Equation (4))


OFA = C12:0 + C14:0 + C16:0(4)


#### 2.3.5. Hypocholesterolemic and Hypercholesterolemic Ratio (H/H)

The H/H ratio is related to the functional activity of fatty acids in the metabolism of lipoproteins regarding plasma cholesterol transport and the risk of cardiovascular disease. The ratio of hypocholesterolemic and hypercholesterolemic fatty acids (H/H) was calculated according to Ivanova and Hadzhinikolova (Equation (5)) [39]. H/H = (C18:1*n*-9 + C18:2*n*-6 + C18:3*n*-3)/(C12:0 + C14:0 + C16:0)(5)

### 2.4. Statistical Analysis

The statistical analyses were calculated using STATISTICA ver. 13.1 software (Statsoft, Kraków, Poland) [40]. The one-way analysis of variance ANOVA (Duncan’s test) was used to test significant differences between the content of fatty acids. The significance level of *p* < 0.05 was used.

## 3. Results 

### 3.1. Fatty Acid Composition and Lipid Quality Indices in the Cheeses and the Cheese-Like Products

The fatty acid composition and the total content of fatty acid groups in the smoked and the unsmoked cheeses and cheese-like products are presented in Table 3. The obtained results indicate that the smoked and the unsmoked cheeses and cheese-like products were characterized by a diversified content of fatty acids. The unsmoked cheeses had a significantly higher total content of saturated fatty acids (SFA) (63.65 ± 1.92%) (*p* < 0.05) compared to the other analyzed products (Table 3). The smoked cheeses had significantly lower total content of SFA but also had a significantly higher (*p* < 0.05) content of short-chain fatty acids (SCFA) (10.07 ± 0.37%). In smoked and unsmoked cheese-like products, total content of both SFA and SCFA were significantly lower (*p* < 0.05). In all cheeses and cheese-like products, in the group of SFA, the palmitic acid (C16:0), the myristic acid (C14:0), and the stearic acid (C18:0) were found in the highest contents (Table 3). The monounsaturated fatty acids (MUFA) and the polyunsaturated fatty acids (PUFA) contents were the highest in unsmoked cheese-like products (39.29 ± 1.49% and 9.13 ± 0.33%, respectively). In other products, the content of these acids was significantly lower (*p* < 0.05). Smoked and unsmoked cheeses were characterized by similar contents of MUFA and PUFA (over 24% and over 2.5%, respectively) (Table 3). The smoked and the unsmoked cheeses had higher contents of branched-chain fatty acids (BCFA) compared to smoked and unsmoked cheese-like products (Table 3). In unsmoked cheese-like products, only C15:0 *iso* and C15: 0 *aiso* were present in amounts of 0.03% and 0.04%, respectively.

In the cheeses and the cheese-like products, the values of lipid quality indices were varied. The content of desirable hypocholesterolemic fatty acids (DFAs) had the highest significance (*p* < 0.05) in the unsmoked cheese-like products (53.15 ± 1.87) (Table 4). In smoked and unsmoked cheeses, DFA had similar values (36.72 ± 0.89 and 36.49 ± 2.17, respectively) (Table 4). Significantly lower values of OFA were found in the smoked and the unsmoked cheese-like products than in the smoked and the unsmoked cheeses. In the presented study (Table 4), the AI value in the unsmoked cheese-like products was the lowest (0.98 ± 0.10). Significantly higher (*p* < 0.05) AI values were found in the other analyzed products. The unsmoked cheese-like products also had the lowest TI value (2.00 ± 0.31). Significantly higher (*p* < 0.05) values of TI were found in the smoked cheese-like products and the smoked and the unsmoked cheeses. In the current study, the lowest values of the hypocholesterolemic/hypercholesterolemic fatty acids ratio were found in the smoked and the unsmoked cheeses (0.41) (Table 4). Significantly higher (*p* < 0.05) values of the H/H ratio were found in smoked and unsmoked cheese-like products (0.90 ± 0.04 and 1.03 ± 0.06, respectively).

### 3.2. Trans Fatty Acids (TFA) in Cheeses and Cheese-Like Products

Figure 2 shows the content of *trans* isomers of C18:1 acid and *trans* isomers of C18:2 acid and cis9*trans*11 C18:2 (CLA) in the cheeses and the cheese-like products. In smoked and unsmoked cheeses, the total content of *trans* C18:1 was at a similar level. A significantly lower (*p* < 0.05) content of these isomers was found in unsmoked and smoked cheese-like products (1.27% and 0.77%, respectively). 

It was found that, in all samples of cheeses in the group of *trans* C18:1, the highest amount of *trans*10 + *trans*11 isomers were found (Table 3). The contents of these isomers in the smoked cheeses ranged from 1.02% to 1.64%, and in the unsmoked cheeses, they ranged from 0.86% to 1.38%. Other marked *trans* isomers of C18:1 in the cheeses were in smaller amounts. In the cheese-like products, the contents of *trans*10 + *trans*11 isomers were lower (from 0.17% to 0.81% in the smoked cheese-like products and from 0.13% to 0.30% in the unsmoked cheese-like products). The conducted research showed that, in the cheese-like products, the contents of marked *trans* isomers were very diverse (Table 3). In the unsmoked cheese-like products, the contents of *trans*6–*trans*9 isomers ranged from 0.05% to 2.92% and in the smoked cheese-like products from 0.10% to 0.20%.

The smoked cheeses were characterized by the highest content of *trans* C18:2 isomers (0.56% of total fatty acids) (Figure 2). Significantly lower (*p* < 0.05) contents of these isomers were found in unsmoked cheeses and smoked and unsmoked cheese-like products. 

The conducted analyses demonstrated differences in *cis*9*trans*11 C18:2 (CLA) content in the cheeses and the cheese-like products (Table 3, Figure 2). No significant differences were found in the content of CLA in smoked and unsmoked cheeses. In the smoked cheeses, the mean content of CLA was the highest (0.49% of total fatty acids), while in the unsmoked cheeses, it was 0.41%. Significantly lower (*p* < 0.05) mean contents of CLA were found in the smoked and the unsmoked cheese-like products (0.18% and 0.06%, respectively).

## 4. Discussion

The profiles of fatty acids in commercial cheeses were determined by various authors such as Rutkowska et al. [41], Zlatonos et al. [42], Grega et al. [43], Talupr et al. [44], Domagała et al. [45], and Paszczyk et al. [46]. In the literature, there is a lack of data on the fatty acid profile of smoked dairy products. According to the literature data, the fatty acid composition in milk fat depends on many different factors such as cow’s diet, season, breed, lactation stage, age, and geographical location [9,47,48,49,50]. The fatty acid profile of dairy products (cheese, butter, fermented milk products) is a result of the raw milk composition and the conditions used in the production process as well as the activity of added starter cultures and ripening time [51,52,53,54]. According to the research by Rutkowska et al. [41] and Zeppa et al. [55], the cheeses from the summer period had a higher UFA content and a lower SFA content compared to cheeses from the winter period. The research showed that the contents of SFA, MUFA, and PUFA in smoked and unsmoked cheeses were at similar levels. In smoked and unsmoked cheese-like products, the fatty acid profiles were varied (Table 3). Partial replacement of milk fat with vegetable fats in cheese-like products caused changes in the fatty acid profile of these products compared to the cheeses. Smoked and unsmoked cheese-like products had lower SFA content and higher MUFA and PUFA contents than cheeses (Table 3). Cheeses with rape oil added analyzed by Ritvanen et al. [56] contained only 7.51% of SFA and were characterized by the higher contents of MUFA and PUFA. In the presented study (Table 3) and the study by Paszczyk et al. [46], the mean contents of MUFA and PUFA in cheese-like products were lower compared to products analyzed by Ritvanen et al. [56].

Milk fat is the main source of SCFA in the diet. Short-chain fatty acids are important from a nutritional point of view, since they possess specific properties associated with important physiological functions. Butyric acid (C4:0) has a beneficial effect on the intestinal flora, has anti-inflammatory activity, and is a factor preventing progression of colorectal cancer and mammary cancer [57,58]. The content of SCFA in the studied products was varied (Table 3). The smoked cheeses were characterized by the highest content of SCFA. Significantly lower (*p* < 0.05) contents of these acids were found in other products. Commercial cheeses analyzed by Paszczyk et al. [46] were characterized by higher content of SCFA and unsmoked cheese-like products by lower content of SFA compared to samples of cheeses and cheese-like products analyzed in this study.

Adequate intakes of both *n*-6 and *n*-3 fatty acids are essential for good health and low risk of cardiovascular disease and type 2 diabetes [59,60,61,62]. Excessive amounts of *n*-6 PUFA and a very high *n*-6/*n*-3 ratio promote the pathogenesis of many diseases, including cardiovascular disease, cancer, and inflammatory and autoimmune diseases, whereas increased levels of *n*-3 PUFA (a lower *n*-6/*n*-3 ratio) exert suppressive effects. Studies indicate that the optimal ratio may vary with the disease under consideration [63,64]. In milk fat, the main PUFA are linoleic acid (*cis*9*cis*12 C18:2, *n*-6) and α-linolenic acid (*cis*9*cis*12*cis*15 C18:3, *n*-3). The contents of these acids in milk fat are about 1.6% and 0.7% of total fatty acid composition, respectively [2]. In smoked and unsmoked cheeses, content of linoleic acid was at a similar level. The content of linoleic acid was lower than in milk (0.38% and 0.31%, respectively) (Table 3). A significantly higher (*p* < 0.05) content of C18:2 acid was found in smoked and unsmoked cheese-like products (7.26 ± 1.08% and 8.80 ± 0.43%, respectively) (Table 3). The content of C18:3 acid in smoked and unsmoked cheese-like products was lower than in cheeses. The higher contents of C18:2 and C18:3 acid in cheeses with rape oil were found by Ritvanen et al. [56]. The AI and the TI indices are of interest to human nutrition, and they are related to the risk of development of cardiovascular diseases [5]. The higher the values of these coefficients are, the higher risk of developing cardiovascular diseases is, because AI indicates the risk of diseases such as atherosclerosis (deposition of fat in the walls of the arteries), and TI determines the possibility of blood clots [39]. In the present study, both AI and TI indices were lower in the smoked and the unsmoked cheese-like products than in cheeses (Table 4).

The most prominent *trans* fatty acids (TFA) in the human diet are monounsaturated fatty acids with 18 carbon atoms. The main TFA in milk fat is *trans* 11 C18:1 (vaccenic acid, VA), and its contents can be influenced by the animal feeding system [65,66]. Under conventional ruminant diets, milk VA percentage is around 40–50% of total C18:1 TFA, whereas *trans*9 and *trans*10 C18:1 are only present in small amounts (5% and 10% on average, respectively) [67,68]. Thus, the consumption of dairy fat would represent a very low intake of *trans*9 and *trans*10 C18:1 isomers, while it is a good source of VA, often described as the “natural TFA”. In contrast, during the industrial hydrogenation of vegetable oils or fats, a wide range of monounsaturated TFA are generated, and the major isomers are *trans*9 C18:1 and *trans*10 C18:1. There is scientific evidence that some fatty acids with a *trans* configuration have an adverse effect on blood lipids and thereby increase the risk of coronary heart disease [69,70]. In analyzed smoked and unsmoked cheeses, *trans* C18:1 isomers were significantly higher than in smoked and unsmoked cheese-like products (Figure 2). The total content of *trans* C18:1 isomers in the commercial cheeses studied by Żegarska et al. [71] was very varied. The content of this group of fatty acids analyzed by Paszczyk et al. [46] in commercial unsmoked cheeses was higher compared to the studied unsmoked cheeses, and in cheese-like products, this content was at a similar level.

The principal natural sources of CLA in the human diet are dairy products from ruminant animals (such as milk, yogurt, and cheese) and meat. The content of CLA in milk ranged from 2 to 23.6 mg CLA/g lipid [72] with *cis*9, *trans*11 C18:2 as the major isomer (75% to 90% of total CLA) [73,74]. The smoked and the unsmoked cheeses were characterized by a higher content of *cis*9*trans*11 C18:2 (CLA) compared to cheese-like products (Figure 2, Table 3). In smoked and unsmoked cheeses, the content of CLA was at a similar level (Table 3). The content of linoleic acid *c*9*t*11 C18:2 (CLA) (0.49% and 0.41%, respectively) was lower than in milk as studied by Månsson [2]. The differences in CLA content in cheese samples may be explained by differences in origin of cheeses, conditions used during the production process (temperature, type of starter cultures, ripening process), and initial CLA content of raw milk used for making cheeses. Lower content of CLA in cheese-like products may be caused by a small proportion of milk fat in the composition of these products. According to Grega et al. [43] and Żegarska et al. [71], the CLA content in commercial cheeses was varied. The authors found a higher content of this acid in cheeses from summer. In commercial cheeses and cheese-like products analyzed by Paszczyk et al. [43], mean contents of CLA were lower compared to the analyzed cheeses and cheese-like products (Table 3).

## 5. Conclusions

The conducted study demonstrated that smoked and unsmoked cheeses and smoked and unsmoked cheese-like products were characterized by various contents of fatty acids and various lipid quality indices.

In smoked cheeses were significantly higher contents of SCFA and significantly lower contents of SFA than in unsmoked cheeses. The contents of MUFA and PUFA in cheeses were at similar levels.

Although the smoked and the unsmoked cheeses contained significantly higher SFA and significantly lower MUFA and PUFA than the smoked and the unsmoked cheese-like products, they appear to be more beneficial to consumer health, because they have a higher level of SCFA, branched fatty acids (BCFA), vaccenic acid (VA), and conjugated linoleic acid *cis*9*trans*11 C18:2 (CLA).

## Figures and Tables

**Figure 1 ijerph-17-00071-f001:**
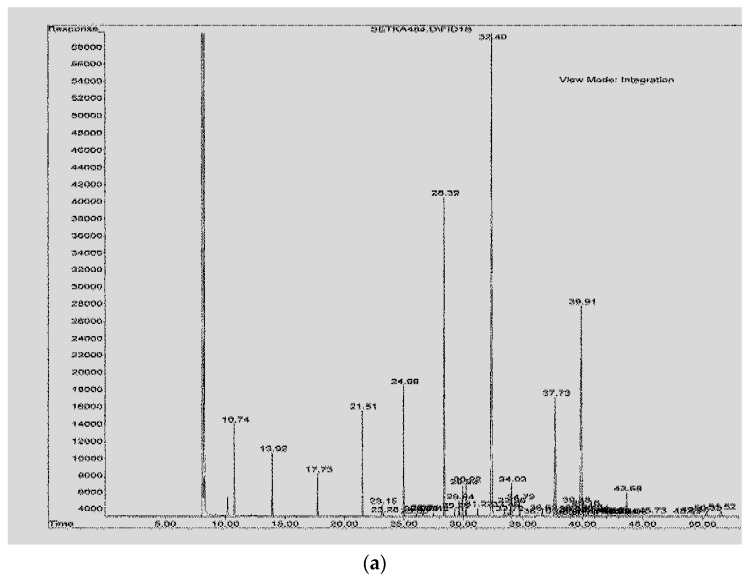
Chromatogram of separation obtained of reference milk fat (**a**) of sample smoked cheese (**b**) and of sample smoked cheese-like products (**c**).

**Figure 2 ijerph-17-00071-f002:**
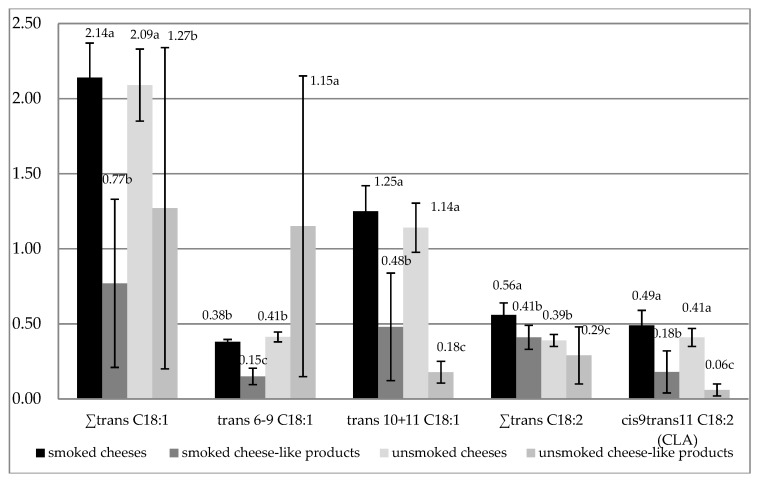
The content of C18:1 and C18:2 *trans* isomers and *cis*9*trans*11 C18:2 (CLA) in products (% of total fatty acids) (mean ± SD); ∑*trans* C18:1—all *trans* isomers of C18:1 (*t*6-*t*9 C18:1, *t*10+*t*11 C18:1, *t*12 C18:1, *t*16 C18:1), ∑*trans* C18:2—all *trans* isomers of C18:2 (*c*9*t*13 C18:2, c9*t*12 C18:2, *t*9*c*12 C18:2, *t*11. C18:2) without *c*9*t*11 C18:2), a,b,c—values denoted by different letters indicate statistically significant differences in the content of *trans* C18:1, C18:2 isomers and *cis*9*trans*11 C18:2 CLA (*p* < 0.05).

**Table 1 ijerph-17-00071-t001:** Analyzed products.

Products	Number of Samples	Producers
Smoked cheese	10	SM Mlekpol (four products); Hochland Polska Sp. z.o.o.; SM Mlekovita (two products); SM Spomlek, OSM Włoszczowa; OSM Sierpc
Smoked cheese-like products	4	SM Mlekovita (two products); OSM Giżycko, OSM Skierniewice
Unsmoked cheeses	10	SM Mlekovita; Polmlek Sp. z.o.o.; OSM Olecko; SM Mlekpol; OSM Sierpc; SM Spomlek; SM Zorina; SP Ryki; SM Włoszczowa; Hochland Polska Sp. z.o.o.
Unsmoked cheese-like products	10	Polmlek Sp. z.o.o. (two products); Pasłęk Sp. z.o.o. (two products); TMT Sp. z.o.o.; SM Mlekovita; PHU JAGR Sp. z.o.o.; PHU Robert Ogorzałek Radom; OSM Włoszczowa; OSM Sierpc

**Table 2 ijerph-17-00071-t002:** Parameters of the chromatographic determination.

Column Temperature	60 °C (for 1 min) to 180 °C, Δt = 5 °C/min
detector temperature	250 °C
injector temperature	225 °C
carrier gas	helium
gas flow	1.5 mL/min
sample injection volume	0.4 μL
split ratio	50:1

**Table 3 ijerph-17-00071-t003:** The fatty acids composition in analyzed products (% of total fatty acids) (Mean ± SD, range).

Fatty Acids	Smoked Cheeses		Smoked Cheese-Like Products		Unsmoked Cheeses		Unsmoked Cheese-Like Products	
n	10		4		10		10	
	Mean ± SD	(Min–Max)	Mean ± SD	(Min–Max)	Mean ± SD	(Min–Max)	Mean ± SD	(Min–Max)
C4:0	3.07 ± 0.13 ^a^	(2.88–3.26)	0.90 ± 0.57 ^c^	(0.37–1.48)	2.20 ± 0.90 ^b^	(0.84–3.23)	0.14 ± 0.11 ^d^	(0.03–0.34)
C6:0	2.29 ± 0.21 ^a^	(2.07–2.83)	0.64 ± 0.39 ^c^	(0.28–1.03)	1.93 ± 0.30 ^b^	(1.45–2.24)	0.10 ± 0.07 ^d^	(0.03–0.22)
C8:0	1.43 ± 0.05 ^a^	(1.34–1.50)	0.43 ± 0.24 ^b^	(0.21–0.66)	1.36 ± 0.08 ^a^	(1.25–1.50)	0.11 ± 0.07 ^c^	(0.06–0.23)
C10:0	3.27 ± 0.10 ^a^	(3.12–3.46	0.94 ± 0.52 ^b^	(0.47–1.40)	3.34 ± 0.26 ^a^	(3.09–3.99)	0.15 ± 0.10 ^c^	(0.09–0.39)
**Σ SCFA ^1^**	**10.07 ± 0.37** ^a^	(9.41–10.51)	**2.92 ± 1.71** ^c^	(1.33–4.57)	**8.82 ± 1.19** ^b^	(7.07–10.06)	**0.31 ± 0.36** ^d^	(0.00–1.18)
C11:0	0.06 ± 0.00 ^a^	(0.06–0.07)	ND		0.07 ± 0.01 ^a^	(0.05–0.09)	ND	
C13:0	0.22 ± 0.01 ^a^	(0.21–0.24)	0.06 ± 0.03 ^c^	(0.03–0.09)	0.13 ± 0.02 ^b^	(0.10–0.15)	ND	
C15:0	1.29 ± 0.06 ^a^	(1.19–1.37)	0.40 ± 0.21 ^b^	(0.21–0.58)	1.26 ± 0.07 ^a^	(1.19–1.39)	0.06 ± 0.05 ^c^	(0.01–0.16)
C17:0	0.57 ± 0.02 ^b^	(0.55–0.59)	0.22 ± 0.08 ^c^	(0.14–0.32)	1.14 ± 0.47 ^a^	(0.70–1.72)	0.11 ± 0.02 ^c^	(0.10–0.15)
C19:0	0.08 ± 0.01 ^b^	(0.07–0.09)	0.03 ± 0.04 ^c^	(0.00–0.08)	0.15 ± 0.02 ^a^	(0.12–0.18)	ND	
**Σ OCFA ^2^**	**2.23 ± 0.09** ^b^	(2.07–2.32)	**0.72 ± 0.36** ^c^	(0.39–1.08)	**2.74 ± 0.49** ^a^	(2.22–3.46)	**0.19 ± 0.06** ^d^	(0.10–0.31)
C13:0 iso	0.12 ± 0.01 ^a^	(0.11–0.12)	0.02 ± 0.03 ^b^	(0.00–0.05)	0.11 ± 0.01 ^a^	(0.09–0.12)	ND	
C14:0 iso	0.15 ± 0.02 ^a^	(0.13–0.18)	0.04 ± 0.02 ^b^	(0.02–0.06)	0.13 ± 0.01 ^a^	(0.12–0.15)	ND	
C15:0 iso	0.28 ± 0.02 ^a^	(0.26–0.33)	0.09 ± 0.04 ^b^	(0.05–0.13)	0.26 ± 0.02 ^a^	(0.23–0.29)	0.03 ± 0.01 ^c^	(0.02–0.04)
C15:0 aiso	0.53 ± 0.02 ^a^	(0.50–0.58)	0.16 ± 0.10 ^b^	(0.07–0.25)	0.53 ± 0.03 ^a^	(0.49–0.59)	0.04 ± 0.02 ^c^	(0.02–0.05)
C16:0 iso	0.30 ± 0.01 ^a^	(0.28–0.32)	0.09 ± 0.06 ^b^	(0.04–0.14)	0.32 ± 0.02 ^a^	(0.29–0.34)	ND	
C17:0 iso	0.17 ± 0.01 ^b^	(0.15–0.18)	0.07 ± 0.05 ^c^	(0.00–0.11)	0.35 ± 0.02 ^a^	(0.32–0.39)	ND	
C17:0 aiso	0.56 ± 0.02 ^a^	(0.53–0.59)	0.13 ± 0.15 ^c^	(0.00–0.27)	0.24 ± 0.09 ^b^	(0.18–0.42)	ND	
**Σ BCFA ^3^**	**2.09 ± 0.08** ^a^	(1.99–2.24)	**0.59 ± 0.43** ^c^	(0.18–1.01)	**1.94 ± 0.12** ^b^	(1.79–2.14)	**0.07 ± 0.04** ^d^	(0.04–0.11)
C12:0	3.80 ± 0.10 ^a^	(3.63–4.01)	1.22±0.53 ^b^	(0.74–1.71)	3.99 ± 0.39 ^a^	(3.55–4.94)	1.03 ± 0.57 ^b^	(0.25–1.74)
C14:0	12.16 ± 0.23 ^a^	(11.73–12.46)	4.10 ± 1.66 ^b^	(2.65–5.68)	12.65 ± 1.03 ^a^	(11.81–15.13)	1.59 ± 0.49 ^c^	(1.29–1.42)
C16:0	32.19 ± 0.97 ^c^	(29.98–33.23)	40.48 ± 3.69 ^b^	(36.60–43.77)	33.07 ± 1.02 ^c^	(32.09–34.85)	43.67 ± 2.08 ^a^	(40.92–46.88)
C18:0	9.01 ± 0.22 ^a^	(8.72–9.34)	5.99 ± 1.22 ^b^	(4.91–7.14)	9.12 ± 0.83 ^a^	(7.48–10.39)	4.73 ± 0.92 ^c^	(4.15–7.24)
C20:0	0.15 ± 0.01 ^c^	(0.14–0.18)	0.28 ± 0.03 ^b^	(0.25–0.30)	0.14 ± 0.02 ^c^	(0.09–0.17)	0.35 ± 0.05 ^a^	(0.31–0.48)
**Σ MCFA+LCFA ^4^**	**57.31 ± 1.25** ^b^	(54.20–58.76)	**52.07 ± 0.72** ^c^	(51.03–52.64)	**58.98 ± 1.84** ^a^	(57.12–62.47)	**50.68 ± 1.47** ^d^	(48.95–52.28)
**Σ SFA ^5^**	**61.63 ± 1.24** ^b^	(58.40–62.90)	**53.38 ± 0.89** ^c^	(52.88–54.62)	**63.65 ± 1.92** ^a^	(61.55–67.89)	**50.59 ± 1.51** ^d^	(48.57–52.44)
C10:1	0.35 ± 0.01 ^a^	(0.33–0.38	0.09 ± 0.06 ^b^	(0.04–0.14)	0.34 ± 0.02 ^a^	(0.31–0.39)	ND	
C12:1	0.04 ± 0.01 ^b^	(0.03–0.04)	ND		0.09 ± 0.01 ^a^	(0.08–0.11)	ND	
C14:1	1.18 ± 0.05 ^a^	(1.11–1.28)	0.29 ± 0.16 ^b^	(0.15–0.43)	1.14 ± 0.08 ^a^	(1.03–1.33)	0.06 ± 0.04 ^c^	(0.03–0.09)
C16:1	1.67 ± 0.05 ^a^	(1.62–1.80)	0.55 ± 0.23 ^c^	(0.35–0.75)	1.12 ± 0.73 ^b^	(0.32–2.16)	0.20 ± 0.04 ^c^	(0.16–0.27)
*t*6-*t*9 C18:1	0.38 ± 0.02 ^b^	(0.36–0.41)	0.15 ± 0.06 ^c^	(0.10–0.20)	0.41 ± 0.03 ^b^	(0.34–0.45)	1.15 ± 1.00 ^a^	(0.05–2.92)
*t*10+*t*11C18:1	1.25 ± 0.17 ^a^	(1.02–1.64)	0.48 ± 0.36 ^b^	(0.17–0.81)	1.14 ± 0.16 ^a^	(0.86–1.38)	0.18 ± 0.07 ^c^	(0.13–0.30)
*t*12 C18:1	0.23 ± 0.02 ^a^	(0.20–0.26)	0.07 ± 0.06 ^b^	(0.02–0.12)	0.27 ± 0.03 ^a^	(0.21–0.32)	ND	
*c*9 C18:1	18.23 ± 0.42 ^c^	(17.33–18.67)	32.57 ± 2.52 ^b^	(30.25–34.79)	18.61 ± 0.81 ^c^	(17.30–20.10)	36.43 ± 1.00 ^a^	(35.04–38.71)
*c*11 C18:1	0.64 ± 0.03 ^c^	(0.59–0.69)	0.98 ± 0.10 ^b^	(0.91–1.12)	0.61 ± 0.05 ^c^	(0.53–0.68)	1.15 ± 0.06 ^a^	(1.07–1.24)
*c*12 C18:1	0.26 ± 0.02 ^a^	(0.22–0.28)	0.12 ± 0.12 ^b^	(0.02–0.26)	0.27 ± 0.03 ^a^	(0.21–0.33)	0.21 ± 0.07 ^a^	(0.14–0.28)
*c*13 C18:1	0.09 ± 0.01 ^a^	(0.08–0.10)	0.05 ± 0.03 ^b^	(0.03–0.09)	0.09 ± 0.01 ^a^	(0.07–0.10)	0.05 ± 0.02 ^b^	(0.03–0.06)
*t*16 C18:1	0.28 ± 0.02 ^a^	(0.24–0.31)	0.08 ± 0.09 ^b^	(0.00–0.18)	0.27 ± 0.03 ^a^	(0.22–0.32)	0.04 ± 0.01 ^b^	(0.03–0.04)
C20:1	0.12 ± 0.02 ^b^	(0.10–0.17)	0.13 ± 0.02 ^a,b^	(0.11–0.16)	0.11 ± 0.01 ^c^	(0.09–0.13)	0.14 ± 0.01 ^a^	(0.13–0.16)
**Σ MUFA ^6^**	**24.78 ± 0.59** ^c^	(24.26–25.78	**35.64 ± 1.32** ^b^	(34.18–36.79)	**24.72 ± 1.29** ^c^	(22.30–26.75)	**39.29 ± 1.49** ^a^	(37.39–41.20)
*c*9*t*13 C18:2	0.19 ± 0.02 ^b^	(0.15–0.24)	0.05 ± 0.03 ^c^	(0.02–0.08)	0.16 ± 0.01 ^b^	(0.15–0.18)	0.34 ± 0.06 ^a^	(0.29–0.45)
*c*9*t*12 C18:2	0.27 ± 0.02 ^a^	(0.24–0.31)	0.22 ± 0.05 ^b^	(0.16–0.26)	ND		ND	
*t*9*c*12 C18:2	ND		ND		0.15 ± 0.02	(0.12–0.17)	ND	
*t*11*c*15 C18:2	0.12 ± 0.06 ^a^	(0.06–0.15)	0.16 ± 0.06 ^a^	(0.09–0.21)	0.08 ± 0.03 ^b^	(0.05–0.11)	ND	
*c*9*c*12 C18:2 *n-6*	1.31 ± 0.12 ^c^	(1.15–1.54)	7.26 ± 1.08 ^b^	(6.11–8.20)	1.53 ± 0.17 ^c^	(1.30–1.86)	8.80 ± 0.43 ^a^	(8.08–9.36)
*c*9*c*12*c*15 C18:3 *n-3*	0.38 ± 0.06 ^a^	(0.29–0.46)	0.23 ± 0.08 ^c^	(0.16–0.30)	0.31 ± 0.06 ^b^	(0.23–0.42)	0.15 ± 0.02 ^d^	(0.12–0.20)
*c*9*t*11 C18:2 (CLA)	0.49 ± 0.10 ^a^	(0.38–0.72)	0.18 ± 0.14 ^b^	(0.07–0.31)	0.41 ± 0.06 ^a^	(0.28–0.48)	0.06 ± 0.05 ^c^	(0.02–0.09)
**Σ PUFA ^7^**	**2.75 ± 0.20** ^c^	(2.45–3.09)	**8.09 ± 0.96** ^b^	(7.02–8.91)	**2.64 ± 0.22** ^c^	(2.15–2.91	**9.13 ± 0.33** ^a^	(8.50–9.50)
**Σ UFA ^8^**	**27.71 ± 0.75** ^c^	(26.72–28.97)	**43.72 ± 2.28** ^b^	(41.20–45.70)	**27.37 ± 1.40** ^c^	(24.45–29.41)	**48.42 ± 1.46** ^a^	(46.38–50.57)

**n**—number of samples, Min—minimum value, Max—maximum value, Mean—mean value, SD—standard deviation, ND—not detected, ^a,b,c,d^—values denoted in rows by different letters indicate statistically significant differences (*p* < 0.05), **^1^ Σ SCFA**—all short-chain fatty acids; **^2^ Σ OCFA**—all odd-chain fatty acids; **^3^**
**Σ BCFA**—all branched-chain fatty acids; **^4^ Σ MCFA+LCFA**—all medium-chain fatty acids + all long-chain fatty acids; **^5^ Σ SFA**—all saturated fatty acids (OCFA, BCFA, MCFA, LCFA); **^6^ Σ MUFA**—all monounsaturated fatty acids; **^7^ Σ PUFA**– all polyunsaturated fatty acids. **^8^ Σ UFA**—all unsaturated fatty acids (Σ MUFA + Σ PUFA).

**Table 4 ijerph-17-00071-t004:** Lipid quality indices in products (Mean ± SD).

Fatty Acids	Smoked Cheeses	Smoked Cheese-Like Products	Unsmoked Cheeses	Unsmoked Cheese-Like Products
n	10	4	10	10
	Mean ± SD	Mean ± SD	Mean ± SD	Mean ± SD
**DFA ^1^**	36.72 ± 0.89 ^c^	49.71 ± 1.15 ^b^	36.49 ± 2.17 ^c^	53.15 ± 1.87 ^a^
**OFA ^2^**	52.68 ± 1.23 ^a^	47.39 ± 1.01 ^b^	54.53 ± 2.51 ^a^	45.86 ± 1.87 ^b^
**AI ^3^**	3.18 ± 0.09 ^a^	1.35 ± 0.16 ^b^	3.31 ± 0.36 ^a^	0.98 ± 0.10 ^c^
**TI ^4^**	3.66 ± 0.14 ^b^	2.28 ± 0.08 ^c^	3.86 ± 0.27 ^a^	2.00 ± 0.13 ^d^
**H/H ^5^**	0.41 ± 0.01 ^c^	0.87 ± 0.05 ^b^	0.41 ± 0.03 ^c^	1.00 ± 0.05 ^a^

**n**—number of samples, Mean—mean value; SD—standard deviation; ^a,b,c,d^—values denoted in rows by different letters indicate statistically significant differences (*p* < 0.05); **^1^ DFA**—hypocholesterolemic fatty acids (ΣUFA + C18:0); **^2^ OFA**—hypercholesterolemic fatty acids (ΣSFA - C18:0), **^3^ AI** (Index of Atherogenicity); **^4^ TI** (Index of Thrombogenicity), **^5^ H/H** (hypocholesterolemic/hypercholesterolemic ratio).

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
