# Peer review of "Fatty Acids Profile, Trans Isomers, and Lipid Quality Indices in Smoked and Unsmoked Cheeses and Cheese-Like Products"

_ijerph, 2019, doi:10.3390/ijerph17010071_

Round 1
Reviewer 1 Report
The Authors decided to evaluate fatty acids profile, with the speciala empasis on trans isomers, of smoked and unsmoked cheeses and cheese-like products obtained from market in Olsztym (Poland). They also calculate same indices to evaluate their quality.
In my opinion this manuscript needs lots of improvements before possible publication and has very low interest to the readers of International Journal of Environmental Research and Public Health in present form. It suist more some journals concernig food quality but after rewrittening and lots of improvements, especially in the essental issues!
There are some major disadvantages of this paper which decrease (or even disqualifies) its qulity:
(1). The number of samples. There were only 34 samples evaluated. According to the Authors they purchased them form local market (Olsztyn, Poland) for two months (which year?). We do not know anything of their origin - where they from local producers? is there something special which distinguish them from other Polish cheeses? Each cheese was analysed in duplicate (sic!!!!) which makes overal number of samples 68, which is rather small for making any distinguishing and conclusions of general meaning. What about the seasonal differences in fatty acids profile of ruminant fat? If the Authors want to make some general statement, they should evaluate much more samples or compare this 34 product (why not 40?) obtained in Olsztyn, Poland from different seasons. It will improve their inference.
(2) In the introduction section the Authors give general and well-known information of fatty acids and cheeses but they do not justify the main purpose of their research. What about other nutritive features of cheeses which also influence their health implications like cholesterol content, odd and branched fatt acids (OBCFA) which are characteristic for ruminant fat, conjugated linoleic acids (CLAs) - which are a group of isomers, not only rumenic acid, other conjugated fatty acids of two and three unsaturated bonds, which are formed in rumen and are present in ruminant fat, like rumenelic acid. The Authors also did not refear to the health implications of smoking procedure and its harmful products, which arise in smoked products. They are of main concern from the consumers point of view. What about the products of lipid peroxidation, which also appear in smoked products (malondialdehyde, oxisterols, oxidised phytosterols)? Are they not important from the consumers pount of view? All of these aspects should be describe in the introduction section.
(3) The Authors did not distinguish natural trans isomers, like vaccenic acid, from trans isomers arrising from plant oils. This is one of the main concerns when cheese and cheese-like products are comparing. This issue should be mentioned in the introduction section and discussed later!
(4) The general idea, that cheese-like products are better, because of higher PUFA content and lower values of calculated indices is not justyfied both from the nutritionl and health point of view.
Author Response
Response to Reviewer 1 Comments
Response 1: As suggested by the Reviewer, the part ,,Materials” has been corrected L175 –L180.
Reviewer: What about the seasonal differences in fatty acids profile of ruminant fat?
Response 2: The aim of the study was not to assess seasonal changes in the fatty acid profile. Research on seasonal changes in cheese fatty acid composition may be an interesting topic for future research.
Response 3: As suggested by the Reviewer, the part “Introduction” has been corrected L86 L167.
Reviewer: The Authors did not distinguish natural trans isomers, like vaccenic acid, from trans isomers arrising from plant oils. This is one of the main concerns when cheese and cheese-like products are comparing. This issue should be mentioned in the introduction section and discussed later!
Response 4: As suggested by the Reviewer, the issue of natural trans isomers and trans isomers formed in vegetable oils has been discussed in the part ,,Introduction" L107 – L131 and in the parts ,,Results" LL534 – L543 and ,,Discussion" L603 –L607.
Point 4: The general idea, that cheese-like products are better, because of higher PUFA content and lower values of calculated indices is not justyfied both from the nutritionl and health point of view.
Response: As suggested by the Reviewer, the part “Conclusions” has been corrected:
The conducted study demonstrated that analyzed smoked and unsmoked cheeses and smoked and unsmoked cheese-like products were characterized by various contents of fatty acids and various lipid quality indexes.
In analyzed smoked cheeses were significantly higher content of SCFA and significantly lower content of SFA than unsmoked cheeses. The content of MUFA and PUFA in analyzed cheeses was at the similar level.
Although smoked and unsmoked cheeses contain significantly higher SFA acids and significantly lower MUFA and PUFA acids than smoked and unsmoked cheese-like products, they appear to be more beneficial to consumer health, because they have a higher level of SCFA, branched fatty acids (BFCA), vaccenic acid (VA) and conjugated linoleic acid cis9trans11 C18: 2 (CLA).
Response: As suggested by the Reviewer, the manuscript was corrected by a professional language editor.
The other remarks noted by the Reviewer have been corrected. Thank you for all the comments.
Reviewer 2 Report
In their paper Paszczyk et al investigated total lipid fatty acids of smoked and unsmoked cheeses and cheese-like products. With respect to the importance of lipid composition and intake for human well-being and health, and the classification of classical and alternative foods, this is a valuable task. However, there are substantial objections against publication in the present form.
Major:
The classification of fatty acids in terms of being "healthy" or "not-healthy" is based on poor evidence, mainly based on old reviews and assumptions. Particularly, starting with ref. 3, there is a Lancet review from 1991, followed by ref 4, a review as well. However, they and the references therein are wrongly cited in terms of the adverse effects of lauric and myristic acid: there is clear evidence that lauric and myristic acid are good guys, and that myristic acid increases the so-called good (HDL-) cholesterol (European Journal of Clinical Nutrition (2003) 57, 735–742). Analytics are not adequate, as fatty acids beyond C18, i.e. arachidonic acid (C20:4n-6=ARA), eicosapentanoic acid (C20:5n-3=EPA), docosatetraenoic acid (C22:4n-6) and docosahexaenoic acid (C22:6n-3=DHA) were not analysed. Hence, any calculation of omega6 to 3 ratios are fiction, similarly those of composition. An indication of FA concentration should be made. The extraction method (line 73ff) and, particularly, washing of the lipid extracts may well decrease the recovery of particularly short chain free fatty acids, a group of lipids abundent in processed food. The preparation of fatty acid methy esters (line 85-86) and the gas chromatography method (line 88ff) are poorly described. A representative chromatogram of standard and sample should be shown. Equations and style of showing the calculation of indexes (line 100ff) are away from adequate style. Abbreviations are repetitively explained (l.113ff; l125ff). This similarly applies to the body of the text. Results and Discussion should be separated. This whole chapter is lengthy and entiring written. The surce of products is not sufficienty described. Are they from different charges or producers or different chees types; are they only locally available; is there an estimate, whether they are similar to products found in other Polish locations or to foreign product? The ingredients acc. to the manufacturer must be declared in a separate table. The text frequently is a repetition of table data, rather then a nice story supplemented by original data shown in tables or figures. The separate discussion should contain the pros and cons of the indexes and equations used. This holds particularly because of lack of evidence from data on the basis of clinical studies. It should noted, whether statements are based on experimental or clinical data (ref. 13 - line 48f), and care must be taken on a balanced use of original literature rather than reviews.Minor:
The word analyzed is redundant after line 71. Everibody expects that the samples were analyzed acc. to the Methods section. This similarly applies to "purchased in Finland". It does not matter, whether they were purchased. In the text values of the table are given, which mostly is superfluous (either or, not both). Also, they are without SD in the text. The style of table 2 is inadequate. Provide mean+/-SD (min;max). Show significance levels. Use an identical caption on top of page 2 of table 2.Legend is poor. Use the same style in Table 3. Put significance levels in greater fond after SD, and explain them in the legend. SFA should be the sum of what's above. Hence, C12:0, C14:0 and C16:0 must be included as a group. Groups should be explained in brackets as FA (like: Cx:0-Cy:0) after abbreviation. Legend is poorly written. Style of Fig. 1: The boxes with data on he bottom are inadequate. Significance levels are not adequately explained. Scaling is not adequate. Nomenclature of CLAs differes from that in the text. CLA is notexplained in the legend of figure. Conclusion is much too long. Actually, it covers half the discussion. Abbreviations have to be clear from the beginning, and not re-explained in the Conclusions. Also, significance levels do not belong in Conclusions. Reference style line 195 and 197 is inadequate Synchronize line spacing of Table 2+3 with Table 1 Synchonize legend of Fig. 1 with table legends.Author Response
Response to Reviewer 2 Comments
Major:
Response: As suggested by the Reviewer, the impact of SFA on our health has been supplemented L86 – L100.
Response: As suggested by the Reviewer, the preparation method of fatty acid methyl esters and the gas chromatography method has been supplemented L195 – L201 and L205 – L209. A representative chromatogram of standard and sample has been added.
Response: As suggested by the Reviewer, equations and style of showing the calculation of indexes has been corrected.
Response: As suggested by the Reviewer, results and discussion are separated and corrected.
Response: As suggested by the Reviewer, the description of the test material has been corrected L175 – L180.
Response: Arachidonic acid, eicosapentanoic acid, docosatetraenoic acid and docosahexaenoic acid were not marked. As suggested by the Reviewer, the n-6 /n-3 ratio was removed.
Minor:
Response: As suggested by the Reviewer, the results have been improved. In the text were average values given are +/-SD.
Response: As suggested by the Reviewer, Table 3 and Table 4 have been corrected, the legend below the Table has been completed.
Response: As suggested by the Reviewer, Figure 2 has been improved.
Response: As suggested by the Reviewer, the conclusions have been corrected L662 – L672.
Response: As suggested by the Reviewer the manuscript was corrected by a professional language editor.
The other remarks noted by the Reviewer have been corrected. Thank you for all the comments.
Reviewer 3 Report
The reason for analyzing smoked vs. non-smoked foods is not mentioned. What is the scientific importance of this comparison?
What is the effect of the smoking process on the fatty acids changes and quality of the mentioned foods?
It has previously been suggested not to name the foods that we consume daily as atherogenic, thrombotic, hypercholesteromiants, carcinogens, etc., (NUTRITION REVIEWS/VOL 46, (9) SEPTEMBER 1988 ) because it is not the food itself that causes these problems but their abuse in the normal diet, coupled with the incorrect lifestyles.
I see of greater scientific interest to identify the possible causes in the preference of the consumer and companies for the production, sale, and consumption of food, and that from the point of view of the authors, are not very healthy.
Author Response
Response to Reviewer 3 Comments
Point 1: The reason for analyzing smoked vs. non-smoked foods is not mentioned. What is the scientific importance of this comparison?
What is the effect of the smoking process on the fatty acids changes and quality of the mentioned foods?
Response: As suggested by the Reviewer, the research objective has been improved L 144 – L167.
Response: As suggested by the Reviewer, in the manuscript text has been corrected. The phrases “food is atherogenic, thrombotic, hypercholesteromiants, carcinogens, etc., has been removed.
Thank you for all the comments.
Round 2
Reviewer 1 Report
I do appreciate all the improvements made by the Authors in their manuscript. However, there are still some points, which need substantial improvement. I do sustain my claiming that the scientific value of this research is rather small (especially regarding the fact the number of samples (34)) and is of faint interest for the general audience. It's hard to believe that 34 samples of smoked and un-smoked cheeses and chees-like products can be a representative for the Polish market. From my point of view this should not be published in the International Journal of Environmental Research and Public Health in present form.
Title: The Authors did not measure the exact content of fatty acids, only their percentage share (profile), so the title is inappropriate.
Intorduction: The Authors have improved this part, which makes it more valuable for the Readers.
Materials: The Authors have presented the origin of their samples, which indicated, that they are not from the area of Olsztyn. Moreover, in the aim of the study the Authors claim their research beeing representative of "the Polish market". Taking into account the number of samples (34 for 4 groups) is too small.
Methods: Why in line 207, in OFA C12:0 is considered as hypercholesterolaemic, whereas in line 218, in H/H is not? Explain reasonably this inconsistency!
Results: It would be much better to group FA in table 3 in SFA, MUFA and PUFA groups, accompying them with sum of SFA etc.
Without proper distinguishing among nautral, valuable trans isomers and industrial trans isomers, the quality of Figure 4 is questionable.
Discussion: The whole part needs in-depth improvement. It is descriptive and only collates the results obtained by Authors with the results of others, without any consideration or an attempt of explanation.
Coclusions: The Authors have completely change their mind about the conclusions of their study. It's an unusall incidene, however present form seems more justified.
Minor remarks:
line 57: trans needs italic
line 73: "this acid..." which? it's not obvious from the context
lines 108-110: this sentence need improvement
line 139: space between 50 and mg
line 150: space between 0.25 and mm
table 2: no space between 250 and 225 and degrees Celsius
line 222: 2.4.7 statistical analysis - why such a number of chapter?
lines 230-254: English language needs improvement; there are some grammatical erros, although the Authors claim, that the manuscript was checked by the professional (....really?)
line: 247: the plural form of "index" is "indices" not "indexes" (sic!)
line 357: sentence is not clear
lines 361-362: "the consumption of...." Was this statement proved by the Authors or maybe it's only their presumption? Explain reasonably and give proper reference!
Author Response
Response II to Reviewer 1 Comments
Point 1 Title: The Authors did not measure the exact content of fatty acids, only their percentage share (profile), so the title is inappropriate.
Response 1: As suggested by the Reviewer, manuscript title has been changed to:
“Fatty Acids Profile, Trans Isomers and Lipid Quality Indexes in Smoked and Unsmoked Cheeses and Cheese-Like Products”
Point 2 Materials: The Authors have presented the origin of their samples, which indicated, that they are not from the area of Olsztyn. Moreover, in the aim of the study the Authors claim their research beeing representative of "the Polish market". Taking into account the number of samples (34 for 4 groups) is too small.
Response: Thank you for this comment. We will take this into account when planning another study. However, we cannot fully agree with this remark. Although the assortment of unsmoked cheeses on the Polish market is large, the assortment of smoked cheeses and smoked and unsmoked cheese-like products is much smaller. Moreover, the availability of these products on the Polish market is very variable. In the period from April to May 2018 on the local market only these types of smoked and unsmoked cheese-like products and smoked cheeses were available. Taking into account that in the literature there is a little information on the fatty acids profile of smoked and unsmoked cheese-like products and smoked cheeses we have decided to carry out the experiment even on such a small amount of different types of cheeses and cheese-like products available on the market.
Point 3 Methods: Why in line 207, in OFA C12:0 is considered as hypercholesterolaemic, whereas in line 218, in H/H is not? Explain reasonably this inconsistency!
Response: Thank you for this valuable attention. The ratio of hypocholesterolemic and hypercholesterolemic fatty acids (H/H) was calculated according to the formula given by Santos-Silva et al. [2002] in which the authors did not include C12:0 acid.
However, the ratio of these acids was calculated again according to the formula presented by Ivanowa and Hadzhinikowa [2015]. The values presented in Table 4 have been corrected.
Point 4 Results: It would be much better to group FA in table 3 in SFA, MUFA and PUFA groups, accompying them with sum of SFA etc.
Response: As suggested by the Reviewer, Table 3 and Table 4 have been changed.
Point 5 Without proper distinguishing among natural, valuable trans isomers and industrial trans isomers, the quality of Figure 4 is questionable.
Response: As suggested by the Reviewer, Figure 2 has been changed.
Point 6 Discussion: The whole part needs in-depth improvement. It is descriptive and only collates the results obtained by Authors with the results of others, without any consideration or an attempt of explanation.
Response: As suggested by the Reviewer, discussion has been changed.
Minor remarks:
Point 7 line 57: trans needs italic
Response: It has been corrected.
Point 8 line 73: "this acid..." which? it's not obvious from the context
Response: As suggested by the Reviewer, has been corrected
Point 9 lines 108-110: this sentence need improvement
Response: As suggested by the Reviewer, this sentence has been corrected.
Point 10 line 139: space between 50 and mg
Response: It has been corrected.
Point 11 line 150: space between 0.25 and mm
Response: It has been corrected.
Point 12 table 2: no space between 250 and 225 and degrees Celsius
Response: It has been corrected.
Point 13 line 222: 2.4.7 statistical analysis - why such a number of chapter?
Response: It has been corrected.
Point 14 lines 230-254: English language needs improvement; there are some grammatical erros, although the Authors claim, that the manuscript was checked by the professional (....really?)
Response: As suggested by the Reviewer, this sentence has been corrected. As suggested by the Reviewer, the manuscript was checked by native speaker.
Point 15 line: 247: the plural form of "index" is "indices" not "indexes" (sic!)
Response: It has been corrected.
Point 16 line 357: sentence is not clear.
Response: As suggested by the Reviewer, this sentence has been corrected.
Point 17 lines 361-362: "the consumption of...." Was this statement proved by the Authors or maybe it's only their presumption? Explain reasonably and give proper reference!
Response: This sentence was written by the authors. It has been deleted.
As suggested by the Reviewer, the manuscript was corrected by native speaker. Thank you for all the comments.
Reviewer 2 Report
The authors have significantly revised their manuscript, and have converted it into an appropriate style. Congrats! There only remain a few changes:
Results in general: remove the word "analyzed" from the text. Everybody is sure that the data are derived from analysis rather than invention or imagination!
Line 401: "demonstrates" rather than "demonstrated", because the analyses have been performed in the past, but the results demonstrate now and for the future, what the facts/conclusions are!
Author Response
Response II to Reviewer 2 Comments
Point 1: Results in general: remove the word "analyzed" from the text. Everybody is sure that the data are derived from analysis rather than invention or imagination!
Response: As suggested by the Reviewer, the word "analyzed" has been removed from the text.
Point 2: Line 401: "demonstrates" rather than "demonstrated", because the analyses have been performed in the past, but the results demonstrate now and for the future, what the facts/conclusions are!
Response: As suggested by the Reviewer, the word "demonstrated" has been changed to "demonstrates".
Thank you for all the comments.
Reviewer 3 Report
I have no further comments. The manuscript has improved substantially and, it is suitable for publication in this journal.
Author Response
Thank you for all the comments.
Round 3
Reviewer 1 Report
I do appreciate all the changes made by the Autors to correct their manuscript. I do recommend to accept this paper, however in table 4 there are still indexes instead of indices!